# Robust Imitation of Diverse Behaviors

**Ziyu Wang**[*]**, Josh Merel**[*]**, Scott Reed, Greg Wayne, Nando de Freitas, Nicolas Heess**
DeepMind
`ziyu,jsmerel,reedscot,gregwayne,nandodefreitas,heess@google.com`

## Abstract

Deep generative models have recently shown great promise in imitation learning for motor control. Given enough data, even supervised approaches can do one-shot imitation learning; however, they are vulnerable to cascading failures when the agent trajectory diverges from the demonstrations. Compared to purely supervised methods, Generative Adversarial Imitation Learning (GAIL) can learn more robust controllers from fewer demonstrations, but is inherently mode-seeking and more difficult to train. In this paper, we show how to combine the favourable aspects of these two approaches. The base of our model is a new type of variational autoencoder on demonstration trajectories that learns semantic policy embeddings. We show that these embeddings can be learned on a 9 DoF Jaco robot arm in reaching tasks, and then smoothly interpolated with a resulting smooth interpolation of reaching behavior. Leveraging these policy representations, we develop a new version of GAIL that (1) is much more robust than the purely-supervised controller, especially with few demonstrations, and (2) avoids mode collapse, capturing many diverse behaviors when GAIL on its own does not. We demonstrate our approach on learning diverse gaits from demonstration on a 2D biped and a 62 DoF 3D humanoid in the MuJoCo physics environment.

## 1   Introduction

Building versatile embodied agents, both in the form of real robots and animated avatars, capable of a wide and diverse set of behaviors is one of the long-standing challenges of AI. State-of-the-art robots cannot compete with the effortless variety and adaptive flexibility of motor behaviors produced by toddlers. Towards addressing this challenge, in this work we combine several deep generative approaches to imitation learning in a way that accentuates their individual strengths and addresses their limitations. The end product of this is a robust neural network policy that can imitate a large and diverse set of behaviors using few training demonstrations.

We first introduce a variational autoencoder (VAE) [15, 26] for supervised imitation, consisting of a bi-directional LSTM [13, 32, 9] encoder mapping demonstration sequences to embedding vectors, and two decoders. The first decoder is a multi-layer perceptron (MLP) policy mapping a trajectory embedding and the current state to a continuous action vector. The second is a dynamics model mapping the embedding and previous state to the present state, while modelling correlations among states with a WaveNet [39]. Experiments with a 9 DoF Jaco robot arm and a 9 DoF 2D biped walker, implemented in the MuJoCo physics engine [38], show that the VAE learns a structured semantic embedding space, which allows for smooth policy interpolation.

While supervised policies that condition on demonstrations (such as our VAE or the recent approach of Duan et al. [6]) are powerful models for one-shot imitation, they require large training datasets in order to work for non-trivial tasks. They also tend to be brittle and fail when the agent diverges too much from the demonstration trajectories. These limitations of supervised learning for imitation, also known as behavioral cloning (BC) [24], are well known [28, 29].

---

[*]Joint First authors.

Recently, Ho and Ermon [12] showed a way to overcome the brittleness of supervised imitation using another type of deep generative model called Generative Adversarial Networks (GANs) [8]. Their technique, called Generative Adversarial Imitation Learning (GAIL) uses reinforcement learning, allowing the agent to interact with the environment during training. GAIL allows one to learn more robust policies with fewer demonstrations, but adversarial training introduces another difficulty called mode collapse [7]. This refers to the tendency of adversarial generative models to cover only a subset of modes of a probability distribution, resulting in a failure to produce adequately diverse samples. This will cause the learned policy to capture only a subset of control behaviors (which can be viewed as modes of a distribution), rather than allocating capacity to cover all modes.

Roughly speaking, VAEs can model diverse behaviors without dropping modes, but do not learn robust policies, while GANs give us robust policies but insufficiently diverse behaviors. In section 3, we show how to engineer an objective function that takes advantage of both GANs and VAEs to obtain robust policies capturing diverse behaviors. In section 4, we show that our combined approach enables us to learn diverse behaviors for a 9 DoF 2D biped and a 62 DoF humanoid, where the VAE policy alone is brittle and GAIL alone does not capture all of the diverse behaviors.

## 2  Background and Related Work

We begin our brief review with generative models. One canonical way of training generative models is to maximize the likelihood of the data: $\max \sum_i \log p_\theta(x_i)$. This is equivalent to minimizing the Kullback-Leibler divergence between the distribution of the data and the model: $D_{KL}(p_{data}(\cdot)||p_\theta(\cdot))$. For highly-expressive generative models, however, optimizing the log-likelihood is often intractable.

One class of highly-expressive yet tractable models are the auto-regressive models which decompose the log likelihood as $\log p(x) = \sum_i \log p_\theta(x_i|x_{<i})$. Auto-regressive models have been highly effective in both image and audio generation [40, 39].

Instead of optimizing the log-likelihood directly, one can introduce a parametric inference model over the latent variables, $q_\phi(z|x)$, and optimize a lower bound of the log-likelihood:

$$\mathbb{E}_{q_\phi(z|x_i)}\left[\log p_\theta(x_i|z)\right] - D_{KL}\left(q_\phi(z|x_i)||p(z)\right) \leq \log p(x). \tag{1}$$

For continuous latent variables, this bound can be optimized efficiently via the re-parameterization trick [15, 26]. This class of models are often referred to as VAEs.

GANs, introduced by Goodfellow et al. [8], have become very popular. GANs use two networks: a generator $G$ and a discriminator $D$. The generator attempts to generate samples that are indistinguishable from real data. The job of the discriminator is then to tell apart the data and the samples, predicting 1 with high probability if the sample is real and 0 otherwise. More precisely, GANs optimize the following objective function

$$\min_G \max_D \mathbb{E}_{p_{data}(x)}\left[\log D(x)\right] + \mathbb{E}_{p(z)}\left[\log(1 - D(G(z)))\right]. \tag{2}$$

Auto-regressive models, VAEs and GANs are all highly effective generative models, but have different trade-offs. GANs were noted for their ability to produce sharp image samples, unlike the blurrier samples from contemporary VAE models [8]. However, unlike VAEs and autoregressive models trained via maximum likelihood, they suffer from the mode collapse problem [7]. Recent work has focused on alleviating mode collapse in image modeling [2, 4, 19, 25, 42, 11, 27], but so far these have not been demonstrated in the control domain. Like GANs, autoregressive models produce sharp and at times realistic image samples [40], but they tend to be slow to sample from and unlike VAEs do not immediately provide a latent vector representation of the data. This is why we used VAEs to learn representations of demonstration trajectories.

We turn our attention to imitation. Imitation is the problem of learning a control policy that mimics a behavior provided via a demonstration. It is natural to view imitation learning from the perspective of generative modeling. However, unlike in image and audio modeling, in imitation the generation process is constrained by the environment and the agent's actions, with observations becoming accessible through interaction. Imitation learning brings its own unique challenges.

In this paper, we assume that we have been provided with demonstrations $\{\tau_i\}_i$ where the $i$-th trajectory of state-action pairs is $\tau_i = \{x_1^i, a_1^i, \cdots, x_{T_i}^i, a_{T_i}^i\}$. These trajectories may have been produced by either an artificial or natural agent.

As in generative modeling, we can easily apply maximum likelihood to imitation learning. For instance, if the dynamics are tractable, we can maximize the likelihood of the states directly: $\max_\theta \sum_i \sum_{t=1}^{T_i} \log p(x_{t+1}^i | x_t^i, \pi_\theta(x_t^i))$. If a model of the dynamics is unavailable, we can instead maximize the likelihood of the actions: $\max_\theta \sum_i \sum_{t=1}^{T_i} \log \pi_\theta(a_t^i | x_t^i)$. The latter approach is what we referred to as behavioral cloning (BC) in the introduction.

When demonstrations are plentiful, BC is effective [24, 30, 6]. Without abundant data, BC is known to be inadequate [28, 29, 12]. The inefficiencies of BC stem from the sequential nature of the problem. When using BC, even the slightest errors in mimicking the demonstration behavior can quickly accumulate as the policy is unrolled. A good policy should correct for mistakes made previously, but for BC to achieve this, the corrective behaviors have to appear frequently in the training data.

GAIL [12] avoids some of the pitfalls of BC by allowing the agent to interact with the environment and learn from these interactions. It constructs a reward function using GANs to measure the similarity between the policy-generated trajectories and the expert trajectories. As in GANs, GAIL adopts the following objective function

$$\min_\theta \max_\psi \mathbb{E}_{\pi_E} \left[ \log D_\psi(x, a) \right] + \mathbb{E}_{\pi_\theta} \left[ \log(1 - D_\psi(x, a)) \right], \tag{3}$$

where $\pi_E$ denotes the expert policy that generated the demonstration trajectories.

To avoid differentiating through the system dynamics, policy gradient algorithms are used to train the policy by maximizing the discounted sum of rewards $r_\psi(x_t, a_t) = -\log(1 - D_\psi(x_t, a_t))$. Maximizing this reward, which may differ from the expert reward, drives $\pi_\theta$ to expert-like regions of the state-action space. In practice, trust region policy optimization (TRPO) is used to stabilize the learning process [31]. GAIL has become a popular choice for imitation learning [16] and there already exist model-based [3] and third-person [36] extensions. Two recent GAIL-based approaches [17, 10] introduce additional reward signals that encourage the policy to make use of latent variables which would correspond to different types of demonstrations after training. These approaches are complementary to ours. Neither paper, however, demonstrates the ability to do one-shot imitation.

The literature on imitation including BC, apprenticeship learning and inverse reinforcement learning is vast. We cannot cover this literature at the level of detail it deserves, and instead refer readers to recent authoritative surveys on the topic [5, 1, 14]. Inspired by recent works, including [12, 36, 6], we focus on taking advantage of the dramatic recent advances in deep generative modelling to learn high-dimensional policies capable of learning a diverse set of behaviors from few demonstrations.

In graphics, a significant effort has been devoted to the design physics controllers that take advantage of motion capture data, or key-frames and other inputs provided by animators [33, 35, 43, 22]. Yet, as pointed out in a recent hierarchical control paper [23], the design of such controllers often requires significant human insight. Our focus is on flexible, general imitation methods.

## 3  A Generative Modeling Approach to Imitating Diverse Behaviors

### 3.1  Behavioral cloning with variational autoencoders suited for control

In this section, we follow a similar approach to Duan et al. [6], but opt for stochastic VAEs as having a distribution $q_\phi(z|x_{1:T})$ to better regularize the latent space.

In our VAE, an encoder maps a demonstration sequence to an embedding vector $z$. Given $z$, we decode both the state and action trajectories as shown in Figure 1. To train the model, we minimize the following loss:

$$\mathcal{L}(\alpha, w, \phi; \tau_i) = -\mathbb{E}_{q_\phi(z|x_{1:T_i}^i)} \left[ \sum_{t=1}^{T_i} \log \pi_\alpha(a_t^i | x_t^i, z) + \log p_w(x_{t+1}^i | x_t^i, z) \right] + D_{KL}\left( q_\phi(z|x_{1:T_i}^i) || p(z) \right)$$

Our encoder $q$ uses a bi-directional LSTM. To produce the final embedding, it calculates the average of all the outputs of the second layer of this LSTM before applying a final linear transformation to generate the mean and standard deviation of an Gaussian. We take one sample from this Gaussian as our demonstration encoding.

The action decoder is an MLP that maps the concatenation of the state and the embedding to the parameters of a Gaussian policy. The state decoder is similar to a conditional WaveNet model [39].

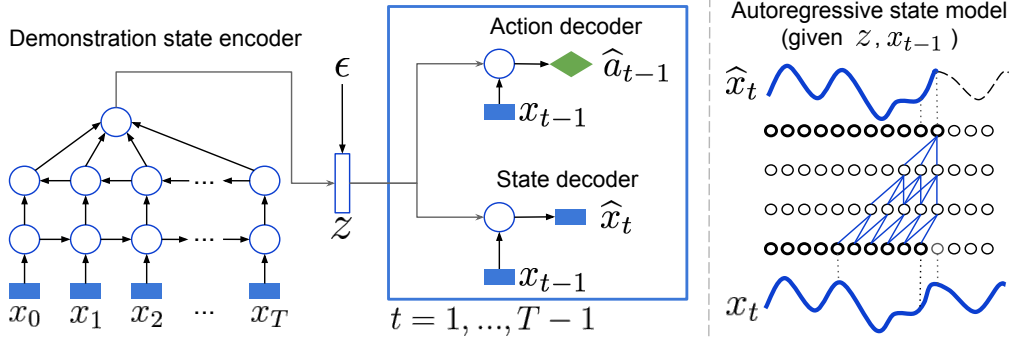

Figure 1: Schematic of the encoder decoder architecture. **LEFT**: Bidirectional LSTM on demonstration states, followed by action and state decoders at each time step. **RIGHT**: State decoder model within a *single* time step, that is autoregressive over the state dimensions.

In particular, it conditions on the embedding $z$ and previous state $x_{t-1}$ to generate the vector $x_t$ autoregressively. That is, the autoregression is over the components of the vector $x_t$. Wavenet lessens the load of the encoder which no longer has to carry information that can be captured by modeling auto-correlations between components of the state vector . Finally, instead of a Softmax, we use a mixture of Gaussians as the output of the WaveNet.

### 3.2 Diverse generative adversarial imitation learning

As pointed out earlier, it is hard for BC policies to mimic experts under environmental perturbations. Our solution to obtain more robust policies from few demonstrations, which are also capable of diverse behaviors, is to build on GAIL. Specifically, to enable GAIL to produce diverse solutions, we condition the discriminator on the embeddings generated by the VAE encoder and integrate out the GAIL objective with respect to the variational posterior $q_\phi(z|x_{1:T})$. Specifically, we train the discriminator by optimizing the following objective

$$\max_\psi \mathbb{E}_{\tau_i \sim \pi_E} \left\{ \mathbb{E}_{q(z|x^i_{1:T_i})} \left[ \frac{1}{T_i} \sum_{t=1}^{T_i} \log D_\psi(x^i_t, a^i_t | z) + \mathbb{E}_{\pi_\theta} \left[ \log(1 - D_\psi(x, a|z)) \right] \right] \right\}. \quad (4)$$

A related work [20] introduces a conditional GAIL objective to learn controllers for multiple behaviors from state trajectories, but the discriminator conditions on an annotated class label, as in conditional GANs [21].

We condition on unlabeled trajectories, which have been passed through a powerful encoder, and hence our approach is capable of one-shot imitation learning. Moreover, the VAE encoder enables us to obtain a continuous latent embedding space where interpolation is possible, as shown in Figure 3.

Since our discriminator is conditional, the reward function is also conditional: $r^t_\psi(x_t, a_t|z) = -\log(1 - D_\psi(x_t, a_t|z))$. We also clip the reward so that it is upper-bounded. Conditioning on $z$ allows us to generate an infinite number of reward functions each of them tailored to imitating a different trajectory. Policy gradients, though mode seeking, will not cause collapse into one particular mode due to the diversity of reward functions.

To better motivate our objective, let us temporarily leave the context of imitation learning and consider the following alternative value function for training GANs

$$\min_G \max_D V(G, D) = \int_y p(y) \int_z q(z|y) \left[ \log D(y|z) + \int_{\hat{y}} G(\hat{y}|z) \log(1 - D(\hat{y}|z)) d\hat{y} \right] dy dz.$$

This function is a simplification of our objective function. Furthermore, it satisfies the following property.

**Lemma 1.** *Assuming that $q$ computes the true posterior distribution that is $q(z|y) = \frac{p(y|z)p(z)}{p(y)}$, then*

$$V(G, D) = \int_z p(z) \left[ \int_y p(y|z) \log D(y|z) dy + \int_{\hat{x}} G(\hat{y}|z) \log(1 - D(\hat{y}|z)) d\hat{y} \right] dz.$$

**Algorithm 1** Diverse generative adversarial imitation learning.

---

**INPUT**: Demonstration trajectories $\{\tau_i\}_i$ and VAE encoder $q$.
**repeat**
    **for** $j \in \{1, \cdots, n\}$ **do**
        Sample trajectory $\tau_j$ from the demonstration set and sample $z_j \sim q(\cdot | x_{1:T_j}^j)$.
        Run policy $\pi_\theta(\cdot | z_j)$ to obtain the trajectory $\hat{\tau}_j$.
    **end for**
    Update policy parameters via TRPO with rewards $r_t^j(x_t^j, a_t^j | z_j) = -\log(1 - D_\psi(x_t^j, a_t^j | z_j))$.
    Update discriminator parameters from $\psi_i$ to $\psi_{i+1}$ with gradient:

$$\nabla_\psi \left\{ \frac{1}{n} \sum_{j=1}^{n} \left[ \frac{1}{T_j} \sum_{t=1}^{T_j} \log D_\psi(x_t^j, a_t^j | z_j) \right] + \left[ \frac{1}{\hat{T}_j} \sum_{t=1}^{\hat{T}_j} \log(1 - D_\psi(\hat{x}_t^j, \hat{a}_t^j | z_j)) \right] \right\}$$

**until** Max iteration or time reached.

---

If we further assume an optimal discriminator [8], the cost optimized by the generator then becomes

$$C(G) = 2 \int_z p(z) JSD \left[ p(\cdot | z) \, || \, G(\cdot | z) \right] dz - \log 4, \tag{5}$$

where $JSD$ stands for the Jensen-Shannon divergence. We know that GANs approximately optimize this divergence, and it is well documented that optimizing it leads to mode seeking behavior [37].

The objective defined in (5) alleviates this problem. Consider an example where $p(x)$ is a mixture of Gaussians and $p(z)$ describes the distribution over the mixture components. In this case, the conditional distribution $p(x|z)$ is not multi-modal, and therefore minimizing the Jensen-Shannon divergence is no longer problematic. In general, if the latent variable $z$ removes most of the ambiguity, we can expect the conditional distributions to be close to uni-modal and therefore our generators to be non-degenerate. In light of this analysis, we would like $q$ to be as close to the posterior as possible and hence our choice of training $q$ with VAEs.

We now turn our attention to some algorithmic considerations. We can use the VAE policy $\pi_\alpha(a_t | x_t, z)$ to accelerate the training of $\pi_\theta(a_t | x_t, z)$. One possible route is to initialize the weights $\theta$ to $\alpha$. However, before the policy behaves reasonably, the noise injected into the policy for exploration (when using stochastic policy gradients) can cause poor initial performance. Instead, we fix $\alpha$ and structure the conditional policy as follows

$$\pi_\theta(\cdot | x, z) = \mathcal{N}\left( \cdot | \mu_\theta(x, z) + \mu_\alpha(x, z), \sigma_\theta(x, z) \right),$$

where $\mu_\alpha$ is the mean of the VAE policy. Finally, the policy parameterized by $\theta$ is optimized with TRPO [31] while holding parameters $\alpha$ fixed, as shown in Algorithm 1.

# 4 Experiments

The primary focus of our experimental evaluation is to demonstrate that the architecture allows learning of robust controllers capable of producing the full spectrum of demonstration behaviors for a diverse range of challenging control problems. We consider three bodies: a 9 DoF robotic arm, a 9 DoF planar walker, and a 62 DoF complex humanoid (56-actuated joint angles, and a freely translating and rotating 3d root joint). While for the reaching task BC is sufficient to obtain a working controller, for the other two problems our full learning procedure is critical.

We analyze the resulting embedding spaces and demonstrate that they exhibit rich and sensible structure that an be exploited for control. Finally, we show that the encoder can be used to capture the gist of novel demonstration trajectories which can then be reproduced by the controller.

All experiments are conducted with the MuJoCo physics engine [38]. For details of the simulation and the experimental setup please see appendix.

## 4.1 Robotic arm reaching
We first demonstrate the effectiveness of our VAE architecture and investigate the nature of the learned embedding space on a reaching task with a simulated Jaco arm. The physical Jaco is a robotics arm developed by Kinova Robotics.

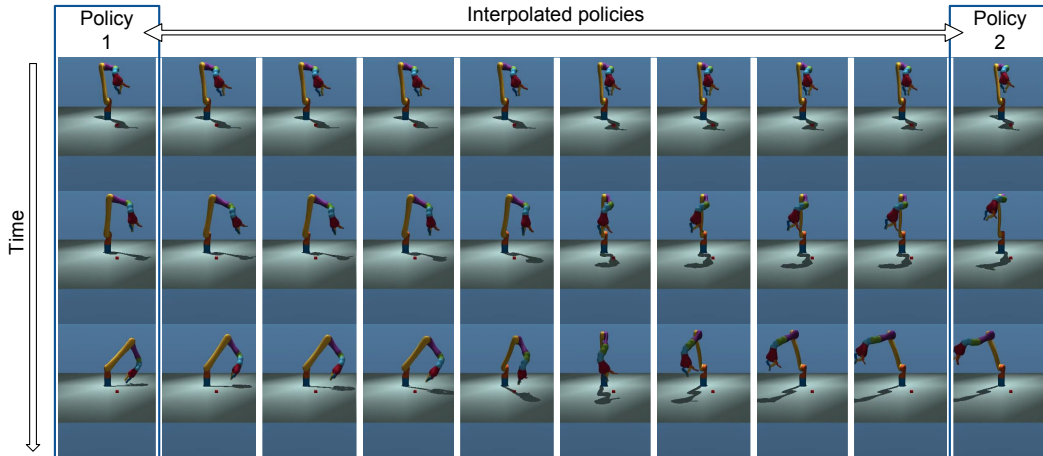

Figure 3: Interpolation in the latent space for the Jaco arm. Each column shows three frames of a target-reach trajectory (time increases across rows). The left and right most columns correspond to the demonstration trajectories in between which we interpolate. Intermediate columns show trajectories generated by our VAE policy conditioned on embeddings which are convex combinations of the embeddings of the demonstration trajectories. Interpolating in the latent space indeed correspond to interpolation in the physical dimensions.

To obtain demonstrations, we trained 60 independent policies to reach to random target locations[2] in the workspace starting from the same initial configuration. We generated 30 trajectories from each of the first 50 policies. These serve as training data for the VAE model (1500 training trajectories in total). The remaining 10 policies were used to generate test data

The reaching task is relatively simple, so with this amount of data the VAE policy is fairly robust. After training, the VAE encodes and reproduces the demonstrations as shown in Figure 2. Representative examples can be found in the video in the supplemental material.

To further investigate the nature of the embedding space we encode two trajectories. Next, we construct the embeddings of interpolating policies by taking convex combinations of the embedding vectors of the two trajectories. We condition the VAE policy on these interpolating embeddings and execute it. The results of this experiment are illustrated with a representative pair in Figure 3. We observe that interpolating in the latent space indeed corresponds to interpolation in task (trajectory endpoint) space, highlighting the semantic meaningfulness of the discovered latent space.

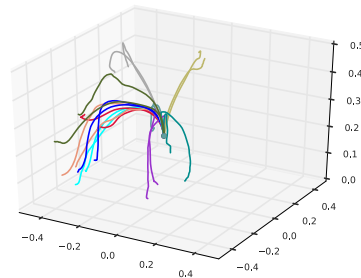

Figure 2: Trajectories for the Jaco arm's end-effector on test set demonstrations. The trajectories produced by the VAE policy and corresponding demonstration are plotted with the same color, illustrating that the policy can imitate well.

## 4.2 2D Walker

We found reaching behavior to be relatively easy to imitate, presumably because it does not involve much physical contact. As a more challenging test we consider bipedal locomotion. We train 60 neural network policies for a 2d walker to serve as demonstrations[3]. These policies are each trained to move at different speeds both forward and backward depending on a label provided as additional input to the policy. Target speeds for training were chosen from a set of four different speeds (m/s): -1, 0, 1, 3. For the distribution of speeds that the trained policies actually achieve see Figure 4, top right). Besides the target speed the reward function imposes few constraints on the behavior. The resulting policies thus form a diverse set with several rather idiosyncratic movement styles. While for most purposes this diversity is undesirable, for the present experiment we consider it a feature.

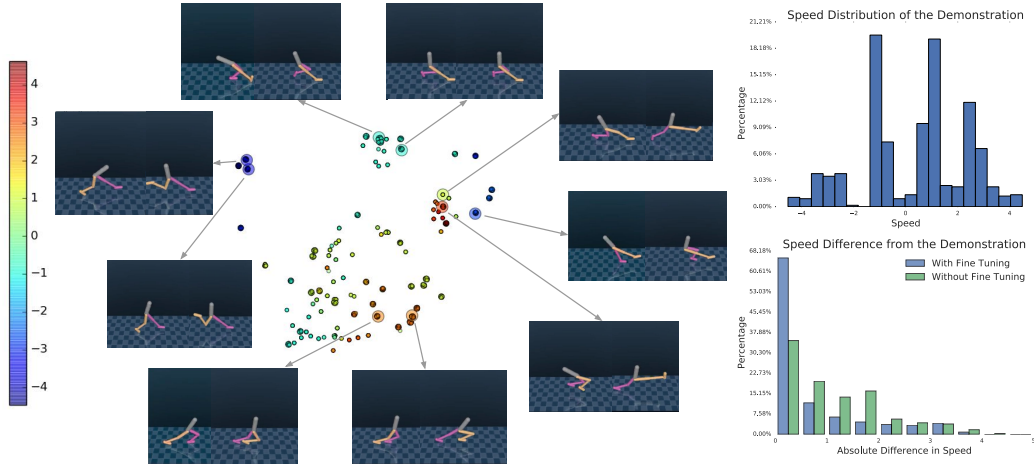

Figure 4: **LEFT**: t-SNE plot of the embedding vectors of the training trajectories; marker color indicates average speed. The plot reveals a clear clustering according to speed. Insets show pairs of frames from selected example trajectories. Trajectories nearby in the plot tend to correspond to similar movement styles even when differing in speed (e.g. see pair of trajectories on the right hand side of plot). **RIGHT, TOP**: Distribution of walker speeds for the demonstration trajectories. **RIGHT, BOTTOM**: Difference in speed between the demonstration and imitation trajectories. Measured against the demonstration trajectories, we observe that the fine-tuned controllers tend to have less difference in speed compared to controllers without fine-tuning.

We trained our model with 20 episodes per policy (1200 demonstration trajectories in total, each with a length of 400 steps or 10s of simulated time). In this experiment our full approach is required: training the VAE with BC alone can imitate some of the trajectories, but it performs poorly in general, presumably because our relatively small training set does not cover the space of trajectories sufficiently densely. On this generated dataset, we also train policies with GAIL using the same architecture and hyper-parameters. Due to the lack of conditioning, GAIL does not reproduce coherently trajectories. Instead, it simply meshes different behaviors together. In addition, the policies trained with GAIL also exhibit dramatically less diversity; see video.

A general problem of adversarial training is that there is no easy way to quantitatively assess the quality of learned models. Here, since we aim to imitate particular demonstration trajectories that were trained to achieve particular target speed(s) we can use the difference between the speed of the demonstration trajectory the trajectory produced by the decoder as a surrogate measure of the quality of the imitation (cf. also [12]).

The general quality of the learned model and the improvement achieved by the adversarial stage of our training procedure are quantified in Fig. 4. We draw 660 trajectories (11 trajectories each for all 60 policies) from the training set, compute the corresponding embedding vectors using the encoder, and use both the VAE policy as well as the improved policy from the adversarial stage to imitate each of the trajectories. We determine the absolute values of the difference between the average speed of the demonstration and the imitation trajectories (measured in $m/s$). As shown in Fig. 4 the adversarial training greatly improves reliability of the controller as well as the ability of the model to accurately match the speed of the demonstration. We also include addition quantitative analysis of our approach using this speed metric in Appendix B. Video of our agent imitating a diverse set of behaviors can be found in the supplemental material.

To assess generalization to novel trajectories we encode and subsequently imitate trajectories not contained in the training set. The supplemental video contains several representative examples, demonstrating that the style of movement is successfully imitated for previously unseen trajectories.

Finally, we analyze the structure of the embedding space. We embed training trajectories and perform dimensionality reduction with t-SNE [41]. The result is shown in Fig. 4. It reveals a clear clustering according to movement speeds thus recovering the nature of the task context for the demonstration trajectories. We further find that trajectories that are nearby in embedding space tend to correspond to similar movement styles even when differing in speed.

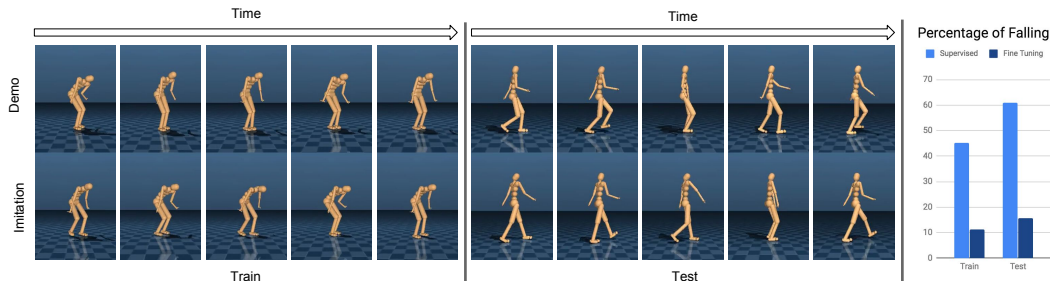

Figure 5: **Left**: examples of the demonstration trajectories in the CMU humanoid domain. The top row shows demonstrations from both the training and test set. The bottom row shows the corresponding imitation. **Right**: Percentage of falling down before the end of the episode with and without fine tuning.

### 4.3 Complex humanoid

We consider a humanoid body of high dimensionality that poses a hard control problem. The construction of this body and associated control policies is described in [20], and is briefly summarized in the appendix (section A.3) for completness. We generate training trajectories with the existing controllers, which can produce instances of one of six different movement styles (see section A.3). Examples of such trajectories are shown in Fig. 5 and in the supplemental video.

The training set consists of 250 random trajectories from 6 different neural network controllers that were trained to match 6 different movement styles from the CMU motion capture data base[4]. Each trajectory is 334 steps or 10s long. We use a second set of 5 controllers from which we generate trajectories for evaluation (3 of these policies were trained on the same movement styles as the policies used for generating training data).

Surprisingly, despite the complexity of the body, supervised learning is quite effective at producing sensible controllers: The VAE policy is reasonably good at imitating the demonstration trajectories, although it lacks the robustness to be practically useful. Adversarial training dramatically improves the stability of the controller. We analyze the improvement quantitatively by computing the percentage of the humanoid falling down before the end of an episode while imitating either training or test policies. The results are summarized in Figure 5 right. The figure further shows sequences of frames of representative demonstration and associated imitation trajectories. Videos of demonstration and imitation behaviors can be found in the supplemental video.

For practical purposes it is desirable to allow the controller to transition from one behavior to another. We test this possibility in an experiment similar to the one for the Jaco arm: We determine the embedding vectors of pairs of demonstration trajectories, start the trajectory by conditioning on the first embedding vector, and then transition from one behavior to the other half-way through the episode by linearly interpolating the embeddings of the two demonstration trajectories over a window of 20 control steps. Although not always successful the learned controller often transitions robustly, despite not having been trained to do so. Representative examples of these transitions can be found in the supplemental video.

## 5 Conclusions

We have proposed an approach for imitation learning that combines the favorable properties of techniques for density modeling with latent variables (VAEs) with those of GAIL. The result is a model that learns, from a moderate number of demonstration trajectories (1) a semantically well structured embedding of behaviors, (2) a corresponding multi-task controller that allows to robustly execute diverse behaviors from this embedding space, as well as (3) an encoder that can map new trajectories into the embedding space and hence allows for one-shot imitation.

Our experimental results demonstrate that our approach can work on a variety of control problems, and that it scales even to very challenging ones such as the control of a simulated humanoid with a large number of degrees of freedoms.

## Footnotes

[2]See appendix for details

[3]See section A.2 in the appendix for details.

[4]See appendix for details.

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
