[Supplementary Material]

Table 1: VAE network specifications.

| Environment | LSTM size | latent size | Action decoder sizes | No. of channels | No. of WaveNet Layers |
|---|---|---|---|---|---|
| Jaco | 500 | 30 | (400, 300, 200) | 32 | 10 |
| Walker | 200 | 20 | (200, 200) | 32 | 6 |
| Humanoid | 500 | 30 | (400, 300, 200) | 32 | 14 |

# A  Details of the experiments

## A.1  Jaco

We trained the random reaching policies with deep deterministic policy gradients (DDPG, [34, 18]) to reach to random positions in the workspace. Simulations were ran for $2.5$ secs or $50$ steps. For more details on the hyper-parameters and network configuration, please refer to Table 1.

## A.2  Walker

The demonstration policies were trained to reach different speeds. Target speeds were chosen from a set of four different speeds (m/s) -1, 0, 1, 3. For each target speed in $\{-1, 0, 1, 3\}$, we trained 12 policies. Another 12 policies are each trained to achieve three target speeds -1, 0, and 1 depending on a context label. Finally 12 policies are each trained to achieve three target speeds -1, 0, and 3 depending on a context label. For each target speed group, a grid search over two parameters are performed: the initial log sigma for the policy and random seeds. We use $4$ initial log sigma values: 0, -1, -2, -3 and three seeds.

For more details on the hyper-parameters and network configurations used see Tables 1 3, and 4.

## A.3  Humanoid

The model of the humanoid body was generated from subject 8 from the CMU database, also used in [20]. It has 56 actuated joint-angles and a freely translating and rotating root. The actions specified for the body correspond to torques of the joint angles.

We generate training trajectories from six different neural network controllers trained to imitate six different movement styles (simple walk, cat style, chicken style, drunk style, and old style). Policies were produced which demonstrate robust, generalized behavior in the style of a simple walk or single motion capture clip from various styles [20]. For evaluation we use a second set of five different policies that had been independently trained on a partially overlapping set of movement styles (drunk style, normal style, old style, sexy-swagger style, strong style).

For more details on the hyper-parameters and network configurations used see Tables 1 3, and 4. To assist the training of the discriminator, we only use a subset of features as inputs to our discriminator following [20]. This subset includes mostly features pertaining to end-effector positions.

Table 2: CMU database motion capture clips per behavior style.

| Type | Subject | clips |
|---|---|---|
| simple walk | 8 | 1-11 |
| cat | 137 | 4 |
| chicken | 137 | 8 |
| drunk | 137 | 16 |
| graceful | 137 | 24 |
| normal | 137 | 29 |
| old | 137 | 33 |
| sexy swagger | 137 | 38 |
| strong | 137 | 42 |

Table 3: Fine tuning phase network specifications.

| Environment | Policy sizes | Discriminator sizes | Critic Network sizes |
|---|---|---|---|
| Walker | (200, 100) | (100, 64) | (200, 100) |
| Humanoid | (300, 200, 100) | (300, 200) | (300, 200, 100) |

Table 4: Fine tuning phase hyper-parameter specifications.

| Environment | Batch sizes per iteration | Initial policy std. | Discriminator learning rate | No. of discriminator update steps |
|---|---|---|---|---|
| Walker | 30000 | $\exp(-1)$ | 1e-4 | 10 |
| Humanoid | 100000 | 0.1 | 1e-4 | 10 |

## B   Data efficiency and hyper-parameter sensitivity

In the case of the 2D walker, we can use the difference in walking speed between the demonstration and the imitation trajectories as a rough measure of performance. Using this measure, we evaluate the data efficiency as well as hyper-parameter sensitivity of the proposed method.

A core idea of our approach is the conditioning of the discriminator on the VAE embeddings of demonstration trajectories. To evaluate gain in data efficiency, we compare our method to that proposed in [20] where the authors condition the discriminator on annotated class labels to achieve diverse behaviors. For the remainder of the section, we refer to this method as *label-conditioned GAIL*.

We also conduct grid searches over the hyper-parameters pertaining to the training of the discriminator to evaluate hyper-parameter sensitivity. We fix the rest of the hyper-parameters to our standard TRPO hyper-parameter settings. To promote fair comparisons, we do not make use of the policies acquired through behavioral cloning (the VAE policies) since label-conditioned GAIL cannot use the same VAE policies without using embeddings. Instead, in all comparison presented in this section, we train policies from scratch. That is we structure our policies to be like:

$$\pi_\theta(\,\cdot\,|x, z) = \mathcal{N}\left(\,\cdot\,|\mu_\theta(x, z), \sigma_\theta(x, z)\right).$$

First, we try to imitate 10 and 60 demonstration trajectories (each from a different policy) using both approaches. The results are summarized in Figure 6 and 7. Our proposed method is not only relatively insensitive to hyper-parameters but also more sample-efficient compared to label-conditioned GAIL.

Interestingly, label-conditioned GAIL is more competitive when imitating 60 behaviors than 10. One possible explanation for this behavior is that imitating more demonstration behaviors promotes sharing between behaviors, therefore enhancing the average performance.

Given the previous observation, the astute reader may wonder how our approach fairs against label-conditioned GAIL when imitating a large number of behaviors. To answer this question, we try to imitate all 1200 walker trajectories. Although these behaviors are generated by 60 distinct policies, many of the policies contain multiple sub-behaviors. Therefore, instead of conditioning label-conditioned GAIL on the indices of policies, we condition it on the indices of the individual trajectories. This choice is far from arbitrary especially when demonstrations are provided by humans in which case it is often difficult to cluster demonstrations into distinct polices. The result is summarized in Figure 8.

When imitating 1200 trajectories, our proposed method performs not dissimilarly when compared to imitating 60 demonstrations. Label-conditioned GAIL, however, fails to learn in 1000 iterations of TRPO updates.

Figure 6: Comparison between our approach and label-conditioned GAIL over 10 demonstration trajectories. The title of each subplot describes the hyper-parameters used. In order, the hyper-parameters are layer sizes, learning rate, and the number of updates of the discriminator per-iteration. Our proposed method is not only relatively insensitive to hyper-parameters but also more sample-efficient compared to label-conditioned GAIL.

Figure 7: Comparison between our approach and label-conditioned GAIL over 60 demonstration trajectories. The title of each subplot describes the hyper-parameters used. In order, the hyper-parameters are layer sizes, learning rate, and the number of updates of the discriminator per-iteration. Our proposed method is not only relatively insensitive to hyper-parameters but also more sample-efficient compared to label-conditioned GAIL.

Figure 8: Comparison between our approach and label-conditioned GAIL over 1200 demonstration trajectories. The title of each subplot describes the hyper-parameters used. In order, the hyper-parameters are layer sizes, learning rate, and the number of updates of the discriminator per-iteration. Our proposed method is not only relatively insensitive to hyper-parameters but also more sample-efficient compared to label-conditioned GAIL. Label-conditioned GAIL in this case, fails to learn.