[Reviews · NeurIPS 2017]

Reviewer 1



The paper proposes a deep-learning-based approach to imitation learning which is sample-efficient and is able to imitate many diverse behaviors. The architecture can be seen as conditional generative adversarial imitation learning (GAIL). The conditioning vector is an embedding of a demonstrated trajectory, provided by a variational autoencoder. This results in one-shot imitation learning: at test time, a new demonstration can be embedded and provided as a conditioning vector to the imitation policy. The authors evaluate the method on several simulated motor control tasks. Detailed comments. Pros: 1) The architecture seems clean and logical, and seems to do the job well. 2) In particular, VAE for trajectory embedding is new compared to recent related approaches. 3) The proposed approach is able to learn complex and diverse behaviors and outperforms both the VAE alone (quantitatively) and GAIL alone (qualitatively). 4) Interpolation between different policies/styles is impressive. Cons: 1) Comparisons to baselines could be more detailed. Currently the comparison to GAIL is purely qualitative, as far as I can see, and only performed on one task. It intuitively seems that GAIL would not perform well, but perhaps it is worth showing clearly in the paper. 2) A discussion of sample efficiency compared to GAIL and VAE would be interesting. What if one trains GAIL per style or target speed - would that work as well as the proposed method? Multi-modality shouldn’t be an issue then. Will VAE work given much more data? 3) Presentation is not always clear, in particular I had hard time figuring out the notation in Section 3. 4) There has been some work on hybrids of VAEs and GANs, which seem worth mentioning when generative models are discussed, like: Autoencoding beyond pixels using a learned similarity metric, Larsen et al., ICML 2016 Generating Images with Perceptual Similarity Metrics based on Deep Networks, Dosovitskiy&Brox. NIPS 2016 These works share the intuition that good coverage of VAEs can be combined with sharp results generated by GANs. 5) Some more extensive analysis of the approach would be interesting. How sensitive is it to hyperparameters? How important is it to use VAE, not usual AE or supervised learning? How difficult will it be for others to apply it to new tasks? 6) A related submission mentioned in lines 124-126 seems quite similar. I have no way to judge to which extent, but seems like it may be worth double-checking. If the work is from the same authors, it can be problematic. Overall, the work is interesting and proposes an elegant and well-performing approach. I think it should be accepted.

Reviewer 2



The paper proposes to combine VAE with GAIL to make the GAIL method more robust by resolving the mode-collapsing problem. For this, the latent feature distribution of demonstration trajectories is learned using VAE and the GAIL objective is modified to be optimized for the expectation over the latent feature distribution. Overall, it becomes solving conditional GAILs where the conditioned embedding is given by VAE encoder and this leads to robust policy learning. The authors claim that, because the learned trajectory feature captures the semantic properties of the demonstration, conditioning on this the resulting generation distribution becomes close to uni-modal. The experiment shows the effectiveness of the proposed method on three continuous control tasks. The paper is well-written and the proposed approach and the experimental results are interesting. I overall enjoyed reading the paper. The followings are some of the questions and comments. Could you elaborate more why q(z|x) is only conditioned on the states x but not along with actions a? And how this is related to the one-shot imitation learning (described in line 127-128)? It could be interesting to see the t-SNE visualization of the trajectory embedding generated by the vanilla GAIL policy? This would provide some additional evidence on the mode-collapsing claim. The proposed method seems helpful in improving sample complexity. Experiment on the varying number of demonstrations or policies would have been interesting to see (e.g., on the speed difference metric).

Reviewer 3



This work deals with the problem of modeling joint action/state trajectory spaces of dynamical systems using a supervised imitation learning paradigm. It approaches the problem by defining a Variational Autoencoder (VAE) that maps the state sequence, via a latent representation, into a (state,action) pair. The approach is validated on three articulated body controller modeling problems: a robotic arm, a 2D biped, and a 3D human body motion modeling. Prior work: the work closely follows (extends) the approach of Generative Adversarial Imitation Learning [11]. The difference here is that, through the use of VAE, the authors claim to have avoided the pitfall of GAN know as mode collapsing. Summary + Addresses a challenging problem of learning complex dynamics controllers / control policies + Well-written introduction / motivation + Appealing qualitative results on the three evaluation problems. Interesting experiments with motion transitioning. - Modeling formulation is somewhat different from GAIL (latent representation) but it rather closely follows GAIL - Many key details are omitted (either on purpose, placed in appendix, or simply absent, like the lack of definitions of terms in the modeling section, details of the planner model, simulation process, or the details of experimental settings) - Experimental evaluation is largely subjective (videos of robotic arm/biped/3D human motion) - Paper appears written in haste or rewritten from a SIGGRAPH like submission - Relies significantly on work presented in an anonymous NIPS submission Detailed comments In general, I like the focus of this paper; designing complex controllers for dynamic systems is an extremely challenging task and the approach proposed here looks reasonable. The use of the VAE is justified and the results, subjectively, are appealing. However, so many details are missing here and some parts of the discussion are unclear or not justified, in my view. For instance, the whole modeling section (3), while not so obscure, still ends up not defining many terms. Eg, what is \pi_E in the first eq of 3.2? (Not numbered, btw) What is \theta in eq3? (also used elsewhere) . What are all the weight terms in the policy model, end of sec 3 (not numbered)? You say you initialize theta to alpha wights, but what does that mean? What is the purpose of Lemma 1? The whole statement following it, l 135-145 sound unclear and contradictory. Eg, you say that eq5 avoids the problem of collapse, yet you state in those lines that is also has the same problem? The decoder models for (x,a) are not explicitly defined. How is the trajectory simulation process actually accomplished? This is somewhat tersely described in l168-172. How are transitions between categories of different motion types modeled as you do not explicitly encode the category class? It is obviously done in the z-space, but is it some linear interpolation? It appears to be (l174-176) but is it timed or instantaneous? What are all the numbers in the Action Decoder Sizes column of Tab 1, appendix?